# Tune in on 11.57 $\mu$ Hz and listen to primary production

Tom J.S. Cox<sup>UA,Y</sup>, Justus E.E. van Beusekom<sup>HZG</sup>, and Karline Soetaert<sup>Y</sup>

<sup>UA</sup>University of Antwerp, Department of Biology, Ecosystem Management research group, Universiteitsplein 1, B-2060 Anwerpen, B <sup>Y</sup>Royal Netherlands Institute of Sea Research (NIOZ) and University of Utrecht, Korringaweg 7, P.O. Box 140, 4400 AC Yerseke, NI

HZGHelmholtz-Zentrum Geesthacht. Institute for Coastal Research, Max-Planck-Strasse 1, 21502 Geesthacht Germany

Correspondence to: Tom Cox (tom.cox@uantwerp.be)

**Abstract.** In this manuscript we present an an elegant approach to reconstruct slowly varying GPP as a function of time, based on  $O_2$  time series. The approach, called complex demodulation, is based on on a direct analogy with amplitude modulated (AM) radio signals. The  $O_2$  concentrations oscillating at the diel frequency (or 11.57  $\mu$ Hz) can be seen as a 'carrier wave', while the time variation in the amplitude of this carrier wave is related to the time varying GPP. The relation follows from an

- analysis in the frequency domain of the governing equation of  $O_2$  dynamics. After the theoretical derivation, we assess the performance of the approach by applying it to 3 artificial  $O_2$  time series, generated with models representative for a well mixed vertical water column, a river and an estuary. These models are forced with hourly observed incident irradiance, resulting in a variblity of GPP on scales from hours to months. The dynamic build-up of algal biomass further increases the seasonality. Complex demodulation allows to reconstruct with great precision time varying GPP of the vertical water column and the river
- model. Surprisingly, it is possible to derive daily averaged GPP complex demodulation thus reconstructs the amplitude of every single diel cycle. Also in estuaries time varying GPP can be reconstructed to a great extent. But there, the influence of the tides prevent achieving the same temporal resolution. In particular, the combination of horizontal  $O_2$  gradients with the O1 and Q1 harmonics in the tides, interferes with the complex demodulation procedure, and introduces spurious amplitude variation that can not be attributed to GPP. But also other tidal harmonics, in casu K1 and P1, introduce diel fluctuations that
- can not be distinguished from GPP. We demonstrate that these spurious effects also occur in real-world time series (Hörnum Tief, De). The spurious fluctuations introduced by O1 and Q1 can be removed to a large extent by increasing the averaging time to 15 days. As such, we demonstrate that a good estimate of the running 15 day average of GPP can be obtained in tidal systems. Apart from the direct merits to estimating GPP from  $O_2$  time series, the analysis in the frequency domain enhances our insights in  $O_2$  dynamics in tidal systems in general, and in the performance of  $O_2$  methods to estimate GPP in particular.

# 20 1 Introduction

Accurate rate estimates of whole ecosystem metabolism are crucial for our understanding of food web dynamics and biogeochemical cycling in aquatic ecosystems. Starting with the seminal work of Odum (1956), time series of in-situ oxygen concentrations have been used to infer rates of gross primary production (GPP) and community respiration (CR) in natural waters (Staehr et al., 2010, 2012). Inferring rates from in-situ  $O_2$  time series requires an accurate description of the dynamics of  $O_2$ , which is affected by a multitude of processes. Most common,  $O_2$  dynamics are described in the time domain. This results in the so-called diel oxygen method, or the Odum method: GPP is estimated from the observed rate of change of oxygen during daylight hours, while adding the rate of community respiration determined at night (Howarth and Michaels, 2000). Recently

we proposed a different approach, based on a description of  $O_2$  dynamics in the frequency domain (Cox et al., 2015). Our analysis resulted in the Fourier method, which estimates time averaged GPP directly from the diel amplitude of  $O_2$  time series.

Both approaches are based on in-situ  $O_2$  concentrations, and this has 2 major advantages over methods that rely on the exsitu incubation of water samples. First 'bottle effects' are avoided: GPP and CR rates are obtained at ambient light fields, and natural levels of turbulence, nutrients and grazing; such conditions are hard to mimic in bottle incubation experiments. Second,

upscaling rates determined by bottle incubation to depth integrated production depends on a range of assumptions about the in-situ light field and mixing of the water body.

Time domain and frequency domain methods differ on a major aspect, namely on how GPP is separated from other processes impacting on  $O_2$  concentrations. For time domain methods, this is a major challenge: the effect of transport processes (turbulent mixing, advective transport, air-water exchange) on the rate of change of oxygen needs to be properly constrained, before

- one can confidently estimate GPP and CR. In open systems with substantial gas exchange, estimates of ecosystem metabolism critically depend on the rate of reaeration, and the resulting GPP and CR values are highly sensitive to reaeration parameterizations (Tobias et al. 2009). Similarly, the diel oxygen method runs into trouble when advection and dispersion processes strongly influence the oxygen concentration, as small errors in the parameterization of these transport processes can lead to order-of-magnitude errors in resulting metabolic rates (Kemp and Boynton, 1980).
- By focussing specifically on the diel harmonic in  $O_2$  time series, frequency domain methods partly circumvent this challenge. As such the Fourier method extends the applicability of in-situ oxygen methods to aquatic systems with a strong imprint of transport processes (Cox et al., 2015). The central idea behind the analysis is that primary production is the dominant process that induces a 24h periodicity in the oxygen concentration while other processes have their imprint at other frequencies. We demonstrated that this assumption is valid in systems with a range of rates of mixing, air–water exchange and primary
- production. However, the method we outlined allowed only a single average GPP estimate representative for average GPP over the period of the  $O_2$  time series was collected. This means that the slow variability of GPP, on time scales ranging from days to years, could not be resolved. This slow variability is expected due to seasonal and weather related variation of light availability, build-up of algal biomass, reduction of algal biomass by grazing, variable discharges and upstream inputs of nutrients and algal biomass. This GPP-variability will show up in  $O_2$ -recordings as slow variations in the amplitude of diurnal  $O_2$  oscillations.
- Moreover, based on simulations we concluded that the performance of the method in estuarine systems is less, due to the tidal movement of an  $O_2$  gradient along the sensor, but we could not explain exactly how this influences the estimate.

In this paper we investigate an elegant approach to reconstruct the slowly varying GPP as a function of time, based on a direct analogy with amplitude modulated (AM) radio signals (Fig 1). AM-radio transmits audible sound (20Hz - 20kHz) by modulating the amplitude of an electromagnetic carrier wave (a single frequency from the 150 kHz - 30 MHz band). In other

words: the sound that we want to hear from our radio is translated to the time varying amplitude of the carrier wave. This is

conceptually similar to a time varying GPP, which causes a time variation in the diel amplitude of oxygen concentrations: the 'carrier wave' then are the  $O_2$  concentrations oscillating with a daily period, thus with frequency f=1/d =11.57  $\mu$ Hz.

# 2 Material and methods

#### 2.1 Theory

5 In (Cox et al., 2015), we demonstrated that the time averaged GPP can be estimated from high frequency in-situ  $O_2$  series as

$$\overline{GPP(t)} \approx 4\pi A_{O_2} \frac{\sin \theta - \theta \cos \theta}{\theta - \frac{1}{2} \sin 2\theta}$$
(1)
with  $\theta = \pi \text{fDL}$ 

where fDL is the relative fraction of light hours during the day, and A<sub>O2</sub> is the amplitude of the diurnal fluctuations in the O<sub>2</sub> series, derived from the Fourier transform as A<sub>O2</sub> = |F(O<sub>2</sub>)(2π)|. We derived this relation by analysing the Fourier transformed differential equations describing oxygen dynamics. It is valid when 4 assumptions are satisfied. First, biochemical processes consuming oxygen have negligible diurnal periodicity, an assumption underlying all diel oxygen methods. Second, the measurement domain is perfectly mixed. This assumption is necessary when one wants to estimate depth averaged GPP from single depth O<sub>2</sub> measurements with any diel oxygen method. Third, diurnal fluctuations of horizontal or vertical fluxes into the measurement domain are negligible. Finally, the evolution of GPP over a day is approximated by a truncated sinusoid.

15 In Cox et al. (2015) we assessed these assumptions and concluded that they are valid in a wide range of systems. In tidal systems the performance of the Fourier method was least, although also there relation 2 holds to a large extent.

When GPP varies slowly with time, so will  $A_{O_2}$ . The mathematical procudure to extract the slowly varying amplitude of a signal is called *complex demodulation* (Bloomfield, 2000). Assume we are interested in the slowly varying amplitude of a cosine function

$$20 \quad x(t) = A(t)\cos(\omega t + \phi)$$

$$= \frac{A(t)}{2}(\exp i(\omega t + \phi) + \exp -i(\omega t + \phi))$$
(2)
(3)

where we used Euler's relation to write the cosine function as a sum of complex exponentials, and with *i* the imaginary unit. The amplitude A(t) is assumed to vary slowly when compared to the periodic oscillation in the cosine functions. Multiplying with  $\exp -i\omega t$  gives

25 
$$y(t) = \frac{A(t)}{2} \exp i\phi + \frac{A(t)}{2} \exp\left(-i(2\omega t + \phi)\right)$$
 (4)

As A(t) is assumed to vary slowly, we can low-pass filter y(t) to get

$$y'(t) = \frac{A(t)}{2} \exp i\phi \tag{5}$$

where the prime denotes low-pass filtering. After noting that for fixed amplitude sinusoid  $|\mathcal{F}(A\cos\omega t)(\omega)| = A/2$ , we have

$$GPP'(t) \approx 4\pi |y'(t)| \frac{\sin \theta - \theta \cos \theta}{\theta - \frac{1}{2} \sin 2\theta}$$
with  $y(t) = O_2(t) \exp 2\pi i t$ 
(6)

This relation allows to estimate slowly varying GPP based on high frequency  $O_2$  time series. It is the central result of this 5 paper.

Diurnal tidal constituents moving back and forth a horizontal oxygen profile will also result in diurnal O<sub>2</sub> fluctuations. While these diurnal constituents are generally much smaller than the semi-diurnal M2 and nearby tidal harmonics, we will demonstrate that they result in diurnal O<sub>2</sub> fluctuations that are not negligible. To assess their impact we will analyse the first order term in the spatial Taylor expansion of simulated oxygen time series (see below for details on simulations). When we
10 denote by O<sub>2</sub>(x,t) the horizontal oxygen profile in a reference frame moving with the tides, the time series recorded by a sensor O<sub>2</sub><sup>S</sup>(t) at a fixed location x = 0 is given by

$$O_2^S(t) = O_2(x(t),t)$$
(7)

$$= O_2(0,t) + \left. \frac{\partial O_2(x,t)}{\partial x} \right|_{x=0} x(t) + \mathcal{O}(x^2(t))$$
(8)

where x(t) is the time varying tidal excursion at the sensor location. The second term in the latter equation represents the first
order effect of the tides moving back and forth the O<sub>2</sub> gradient. Based on simulation results, we can calculate this first order correction for the impact of the tides on the GPP estimate, by substituting O<sub>2</sub>(t) in equation 6 by O<sub>2</sub><sup>S</sup>(t) - \frac{\partial O\_2(x,t)}{\partial x} \Big|\_{x=0} x(t). Obviously, in real world situation such correction term can not be easily estimated and would require an estimate of time varying horizontal gradients and tidal excursion at the sensor location. But when numerically simulating oxygen concentration, this additional information is readily available. Similarly, it is straightforward to apply the classic Odum method, sensu Cole et al.
(2000), to the model output for comparison.

### 2.2 Application to artificial data

To assess the performance of complex demodulation to estimate time varying GPP, we use artificial data sets generated with two numerical models. The first model describes a water body with no appreciable lateral transport of oxygen, representative for a lake or the surface layer of the ocean, where vertical turbulence and air-water exchange are the dominant transport processes. The second describes a typical riverine or estuarine situation, characterized by substantial horizontal gradients in

25

the  $O_2$  concentration. A full description of these models is found in (Cox et al., 2015).

As the models are forced with observed hourly irradiance data, GPP is a function of time. This causes an overall seasonality in GPP as well as shorter term variability due to changes in cloudiness. As forcing we used incident light recorded in 2009 on the roof of NIOZ-Yerseke (Nl.) using a Licor LI-190 SA cosine sensor. Additionally, the dynamic build-up and break-down of

30 algal biomass add both to the seasonality and to the short term variability.

We use a single numerical model to simulate typical riverine and estuarine situations. To emulate the occurence of tides, the output of the riverine model is resampled, using simulated velocities generated with a separate 1D tide resolved hydrodynamic model. We thus simulate estuarine transport with the riverine model, assuming a reference frame moving with the tides. This allows us to investigate the influence of tides on the GPP estimates (more details in Cox et al. (2015)).

### 5 2.3 Application to real world data

To test the performance on real world data, we used a full year (2008) of  $O_2$  measurements, recorded at the Hörnum Tief measurement pole south of the island of Sylt in the German Wadden Sea. The  $O_2$  time series were quality checked: outliers and other unphysical observations were removed manually (Götz Floeser, personal communication). There were short gaps in the time series (< 4h) that were linearly interpolated to obtain a consistent data set at a fixed sampling rate ( $\Delta t$ =10 min).

- 10 Unfortunately, no reference data of GPP in the same year as the  $O_2$  time series is available. Therefore we compare the results to GPP estimates from bottle incubations, obtained at the nearby List tidal basin in 2004 (Methods and site description, see Loebl et al. (2007)). Surface specific GPP rates are divided by the basin averaged water depth (2.7m) to convert to volume specific rates. Since there is no a priori guarantee that rates of primary production are similar in 2004 and 2008, we use a time series of Chl a observations to assess inter-annual variability in spring Phaeocystis blooms. The List Deep was sampled twice
- 15 per week in the main tidal channel near the entrance of the List tidal basin. A second shallower station at the entrance of a small tidal bay (Königshafen) was sampled when tides permitted. With this temporal resolution, bloom events are always captured (e.g. Loebl et al. (2007); van Beusekom et al. (2009). As an indicator of bloom intensity we calculated the mean of the four highest values observed in May, representing about one third of the observations in that month.

### **3** Results

### 20 3.1 Water column and river model

The results of the simulations with the open water model are shown in Fig. 2. The top panel shows the simulated depth averaged  $O_2$  concentrations over the course of the year. The bottom panel shows both simulated depth averaged GPP and reconstructed GPP from complex demodulating the  $O_2$  signal. Simulated GPP was filtered using a moving average filter of 1 day width - the values in Fig. 2 thus represent daily averaged GPP. The correlation between simulated and reconstructed GPP is very strong  $(r^2 = 0.995)$ . The possibility to reconstruct daily averaged GPP from  $O_2$  series might come as a surprise, since this effectively means that the amplitude of the diel oxygen fluctuations were resolved up to every single cycle. This is part of the power of complex demodulation. Very similar results are obtained with the riverine model ( $r^2 = 0.997$ , model output not shown). These results reaffirm that under the simulated conditions, air-water exchange and in horizontal dispersive transport have negligible diurnal contributions, and thus need not be quantified when estimating GPP from the diurnal amplitudes of  $O_2$  time series. In

30 Cox et al. (2015) we demonstrated that this allows the application of the classic Fourier method to a range of real-world systems. This result can thus be generalized to complex demodulation to estimate time varying GPP. Important at this point is

to stress the difference with Fig 3. and 6. in (Cox et al., 2015), where 14 day moving averages of GPP were compared to results from the Fourier method applied to 14 day subsets of  $O_2$  series. The fine resolution obtained with complex demodulation here would not be possible with the classical Fourier method.

# 3.2 Estuarine model and the nature of first order contribution of advective transport

5 When applied to simulated O<sub>2</sub> series with the estuary model, complex demodulation still nicely captures the seasonality in daily GPP-rates, and also some of the variability on shorter scales (Fig. 3). But large mismatches are apparent, particularly during Winter and early Spring when GPP is low. These mismatches can be largely explained by the first order effect of the advective tidal transport. Indeed, including the first order term in the spatial Taylor expansion (see methods section), strongly improves the GPP estimate (Fig. 5, top panel). The short episodes of overestimation that are still present during Winter time, illustrate that the first order term is mostly dominant but higher order terms in the expansion are sometimes important.

# 3.2.1 Close to diurnals in tidal excursion

Given the dominant contribution of the first order term, we focus on its nature for better understanding of the impact of advective transport on  $O_2$  concentrations. Recall that this term is the product of the  $O_2$  gradient and the tidal excursion (Eq. 8) and that interference with GPP estimation means that (close to) diurnal constituents of this term are important.

- 15 If the horizontal  $O_2$  gradient is constant or slowly varying, the harmonics in the first order term will be those present in the tidal excursion x(t). The most dominant harmonics in tidal velocity (and thus in tidal excursion) are the lunar diurnals O1 (T=25.8193 hours, Amplitude=0.0284 m  $s^{-1}$ ) and K1 (T=23.9344 hours, Amplitude=0.0162 m  $s^{-1}$ ), the solar diurnal P1 (T=24.0659 hours, Amplitude=0.0134 m  $s^{-1}$ ) and the large lunar elliptic diurnal Q1 (T=26.868350 hours, Amplitude=0.0098 m  $s^{-1}$ ) (Fig. 4). Although these components are very small compared to the major M2 component (Amplitude=1.18 m  $s^{-1}$ )
- 20 and other components with about semi-diurnal periodicity, they are the ones that interfere with the diurnal fluctuations we want

to attribute to primary production. How those close to diurnals show up in  $O_2$  time series is elucidated by looking at how they are propagated in the complex

demodulation procedure. The multiplication of the time series with  $\exp(-i\omega t)$  has the effect of shifting all frequencies in the signal down with  $\omega$ . As a result, any frequency close to the modulation frequency  $\omega_1$  will lead to low frequency fluctuations 25 in the demodulated signal. Taking O1 as an example,  $f = |f_1 - f_{O1}|$  corresponds to a period of 14.2 days. If the low pass filter in the second step of complex demodulation is too wide, these low frequencies are not filtered out. This is the case in all results presented so far, all obtained with a moving average filter of 1 day. Consequently, the spectrum of the correction term

contains down-shifted frequencies corresponding to the dominant harmonics (Fig. 5, central panel) - a low frequency peak at  $|f_{K1} - f_{P1}|/2$  and two peaks at larger frequencies  $|f_1 - f_{O1}|$  and  $|f_1 - f_{Q1}|$ . The low frequency peak is a special case: it results

from the interaction of the K1 and P1 terms. Indeed, summing two cosine waves with slightly different frequencies  $f_1$  and  $f_2$  results in a signal with frequency |f1 + f2|/2 with a slowly varying amplitude, a phenomenon known as "beat". If both waves have equal amplitude, the frequency of this slow amplitude variation is given by |f1 - f2|/2 (when the amplitudes are not equal, the frequency becomes a function of time). Thus, approximately the interaction between K1 and P1 results in a signal

with very close to diurnal periodicity (T 0.9999985 days) and with an amplitude that evolves periodically in time with a period of 365.1 days. This K1, P1 interaction impact on  $O_2$ -concentrations is so close to  $\omega_1$  that its effect is indistinguishable from fluctuations caused by GPP. Unless we can constrain this interaction with additional measurements or numerical simulations, this puts an inherent limit on the use of in-situ O2 time series for GPP estimation.

- 5
- In contrast, the impact of O1 and Q1 can be largely removed simply by decreasing the filter width (i.e. increasing the averaging time). Indeed, using a moving average filter of 15 days has almost the same effect as applying the correction term (Fig. 5, bottom panel). Obviously, this also means that GPP variability on shorter than 15 day time scales can not be resolved. But the correspondence between complex demodulated  $O_2$  and the 15d moving average GPP is striking. This is an important finding since in real world situations it would be very hard to measure the correction term. This would require the time resolved
- observation of both the  $O_2$  gradient and the tidal excursion at the measurement location. In contrast, increasing the averaging 10 time is straightforward.

We can't stress enough that the impact of tidal harmonics on  $O_2$  is a physical process (advective transport of the  $O_2$  gradient) and not a characteristic of the method. At this point it is relevant to take a look at the results of the classic Odum method. Applying the Odum method to the tidal  $O_2$  time series to estimate daily GPP results in huge fluctuations (Fig. 6, top panel).

Surprisingly, these fluctuations have exactly the |f1 - fO1| frequency, as can be seen from the spectrum (Fig. 6, bottom panel). 15 On the same spectrum we see that also the |f1 - fQ| frequency is present. And indeed, 15d averaging the daily GPP estimates removes most of the fluctuations. Still, as expected for tidal systems, the performance of the Odum method is rather poor.

### 3.2.2 Diurnals in O<sub>2</sub> gradient

The small difference that is still present between the first order corrected and the 15d filtered complex demodulation has multiple potential causes (Fig 5, bottom). First and foremost this is the result of the K1, P1 interaction that can't be filtered 20 out (see above). Second are the diurnal fluctuation in the  $O_2$ -gradients. However, because of the product with tidal excursion, all but the lowest tidal excursion frequencies will result in a diurnal contributions to  $O_2$ . As a result, this effect will be small. Similarly, all frequency couples with a difference of  $\omega_1$  will have a diurnal contribution in the product, but these will average out to (almost) zero. It followed already from simulation with the river model and in (Cox et al., 2015), that the effect of turbulent dispersion of such gradient had a neglible effect on diurnal  $O_2$  fluctuations. Combined, the impact of diurnals in 25 O<sub>2</sub>-gradient will generally be small. Thus it is no surprise that the harmonics in the tidal excursion dominate the correction term, as shown in Fig 5 (central panel).

#### 3.3 Real world data

Applying the complex demodulation procedure to real world  $O_2$  series shows the same phenomena as above. As an example, we analyse an  $O_2$  time series from the Hörnum Tief measurement pole near Sylt island in the German Wadden Sea. At this 30 location, the impact of the O1 component is even larger than in our simulations. This is both obvious from the demodulated time series and from the spectrum of the difference between the 1d and 15d low pass filtered demodulated  $O_2$  series (Fig. 7, left panels). Compared to the simulation results, however, the impact is much less apparent in winter. This follows from the near absence of horizontal gradients in  $O_2$  during Winter time; therefore tidal  $O_2$  oscillations are very small. The GPP estimates from bottle incubations in the nearby List basin provide further evidence that the oscillations in the 1-day filtered GPP estimate are spurious. Indeed, the spectrum of the residuals does not show any of the typical tidally induced peaks (Fig. 7, right panels).

Inter-annual differences in phytoplankton bloom dynamics hamper a comparison of the 2004 and 2008 estimates of GPP

- rates (compare van Beusekom et al. (2009)). 2004 was characterized by a large Phaeocystis bloom when compared to the following years. The peak Chl a concentration (mean of four highest values) in May was 21.1 mg L<sup>-1</sup> in 2004, compared to 16.1, 15.1 18.2 and 12.8 respectively in 2005-2008. Peak biomass in 2008 was thus estimated to be mearly 61% of peak biomass in 2004. Therefore we believe that the difference in estimated peak GPP (107  $\mu$ molO<sub>2</sub>L<sup>-1</sup>d<sup>-1</sup> in 2004 versus 53  $\mu$ molO<sub>2</sub>L<sup>-1</sup>d<sup>-1</sup> in 2008) reflects real differences in GPP rates, rather than a difference between methodologies. This is further
- supported by the good correspondence concerning the onset of the bloom (beginning of April), the total duration of productive period (April-September), and the height of GPP rate during Summer (Fig. 7, top panels). However, this interpretation rests on several assumptions, which will be discussed below.

# 4 Discussion

# 4.1 Advantages of complex demodulation

- The complex demodulation procedure presented here is an expansion of the Fourier method. The major benefit over the approach in (Cox et al., 2015) is that complex demodulation gives a theoretically consistent framework to deal with time-varying amplitudes, and thus time-varying GPP. The theoretical derivation of the Fourier method in Cox et al. (2015) in principle relies on O2-fluctuations with constant diurnal amplitude. The pragmatic approach to deal with time varying GPP was to apply the results on moving windows, but this is inconsistent with the basic assumption of constant diurnal amplitude. The framework
- of complex demodulation solves this inconsistency. As a surprising result, we find that (in non-tidal systems) the temporal resolution is very fine: daily values of GPP estimate can be estimated. This would not be possible with the classical Fourier method.

A second advantage is that this theoretical framework allows to understand and analyse the impact of different tidal harmonics on  $O_2$  time series in tidal systems. In Cox et al. (2015) we concluded that "Neglecting diel fluctuations in advective

- tidal fluxes is by far the largest source of [...] error on the Fourier method". Therefore, the improved insight in the impact of advective transport on  $O_2$  concentrations is an important progress. From the study of the behaviour of the first order spatial Taylor expansion, we have shown that close to diurnal tidal harmonics show up in GPP estimates. Our results show that, while close to diurnals are much smaller in magnitude than the semi-diurnals, they have a strong and clearly identifiable imprint in the  $O_2$  series. This explains why the GPP estimates, when calculated with a 1 day filter, are fluctuating with ~15 day period
- (Fig. 7). These fluctuations are spurious and do not reflect real GPP fluctuations. It is important to acknowledge that this is not a property of the method, but a result of advective transport of an  $O_2$  gradient. This is further exemplified by the appearance of the same harmonics in the application of classic Odum approach.

We have shown that this spurious GPP can be largely filtered out with a 15d moving average filter. This comes at the expense of the resolution of the GPP estimate: variability on time scales smaller than 15 days can not be resolved. Resolving variability on time scales smaller than 15 days will be hard, perhaps impossible, in tidal systems with significant diurnals. This boils down to measuring or estimating at least the first order spatial correction term, which consists of two factors: the tidal excursion and

5 the  $O_2$  gradient. It might be possible to estimate the tidal excursion based on hydrodynamic models. Since those models are not typically built to correctly reproduce the diurnals, some care has to be taken. More difficult is the  $O_2$  gradient. Indeed, the gradient that appears in equation 8 is the horizontal gradient in the reference frame moving with the tides. It is not immediately clear if and how this can be calculated from observed  $O_2$  time series, but it would require a number of  $O_2$  sensors along the horizontal.

### 10 4.2 Application to real-world data

15

The choice of Hörnum Tief as a case study was motivated by the availability of a long term, quasi-continuous time series. Although high frequency  $O_2$  measurements are increasingly available, acquiring uninterrupted long-term time series is still a challenge. Regular maintenance is a necessity, typically on a bi-weekly or monthly basis, to keep sensor tips free from biofouling, and to prevent memory and battery from running out. To avoid data gaps during maintenance, often the only solution the alternate deployment of multiple sensors and loggers. But even this does not preclude the occurrence of accidental

- system breakdown. Nevertheless, guaranteeing uninterrupted  $O_2$  time series requires less effort than all other procedures to obtain year long daily averaged GPP estimates. This makes in-situ  $O_2$  methods so attractive for GPP estimation.
- The purpose of including a real-world time series was to demonstrate that the theoretically predicted imprint of tidal diurnals are easily identified in real systems and can be very large. As a side result, the GPP rate estimates seem to correspond well with reference data from bottle incubations, apart from the period of the *Phaeocystis* bloom (Fig 7). This is not a trivial result. The derivation of the fundamental equation relating the diurnal amplitude of O<sub>2</sub> concentrations to GPP (Eq. 2), rests on the assumption that the measured O<sub>2</sub> concentrations are representative for the average concentrations in a certain volume. The measurement pole is located in Hörnum Tief, the major tidal channel through which the tidal basin fills and empties. So the question is: for which volume are the diurnal oscillations in O<sub>2</sub> concentrations representative? Are these fluctuations representative for the whole tidal basin, or for (part of) the tidal channel only? Put differently: what is the variability in diurnal O<sub>2</sub>-amplitudes over the whole tidal basin? There is no data to verify this. Here we assumed that the impact of GPP on O<sub>2</sub> concentrations is effectively smeared out over the whole basin, by horizontal and vertical mixing. Consequently, to compare with rates derived from bottle incubations, we assumed that there is no spatial variability in diurnal amplitudes, and thus that
- 30 conversion of surface specific GPP rates from List tidal basin to volume specific rates by dividing with the average basin depth, rests on the same assumption. Additionally we assumed that basin averaged production in List and Hörnum tidal basin are equal. These are two strong assumptions that need further examination. The comparison of the productivity in both basins, and of the spatial variability of diurnal  $O_2$  fluctuations would require addditional data and/or numerical simulations with reactivetransport models that are specifically set-up for those 2 basins. This is beyond the scope of the current study.

GPP rates derived from complex demodulation of Hörnum Tief  $O_2$  data is representative for GPP in the whole basin. The

A more detailed study to the applicability of Fourier methods to tidal basins such as Hörnum, would also allow to assess another crucial assumption underlying Eq. 2, namely that the impact of other processes with diurnal periodicity have negligible impact on diurnal  $O_2$  amplitudes. In Cox et al. (2015) we demonstrated that this assumption holds for a range of real-world systems, although in some cases a significant upward or downward bias can be expected. Those results are also applicable here:

- in situations where the Fourier method will work, estimating time varying GPP with complex demodulation will also work. The bias that is present in certain circumstances is determined by the light saturation parameter of photosynthesis, the vertical turbulent mixing rate, light extinction coefficient, the water depth and the piston velocity. In most realistic settings, this bias is less than 10%. In some specific settings it will be larger: this includes shallow systems where air-water exchange has large effect on concentrations, deep systems where vertical mixing is not fast enough. When vertical mixing is low, the measurement
- depth is of crucial importance when only a single sensor is available. Additionally piston velocities can be affected by regular coastal winds and currents when e.g. systematic local winds at dawn and sunset would prevail.

Diurnal fluctuations in  $O_2$  concentrations will always result in corresponding diurnals in air-water flux, even with constant piston velocities. This induces a dampening in the  $O_2$  fluctuations, and consequently a downward bias in GPP estimates when this is not taken into account. In a vertically mixed water column, the factor that describes this dampening can be estimated as

$$15 \quad \sqrt{1 + \left(\frac{k}{\omega_1 d}\right)^2} \tag{9}$$

with k the piston velocity, d the depth of the water column and  $\omega_1$  the diurnal frequency (Cox et al., 2015). Assuming a large of  $k = 7 \text{ms}^{-1}$ , and with a basin averaged depth of about 2.5m, we estimate a dampening of about 10%. Neglecting this dampening, as we did in the comparison with bottle incubations, would results in downward biassed GPP estimates. Unfortunately, there exist no analytical expressions to estimate diurnal components in other processes (e.g. vertical mixing). Therefore, they can only be estimated with a numerical model study that is out of the scope here.

To conclude, the correspondence with results from bottle incubations in the nearby List tidal basin is encouraging but not conclusive and a more detailed numerical study is required.

# 5 Conclusions

- Assessing the accuracy of the GPP rate estimates still needs a case by case approach to quantify the different types of bias. But 25 Nevertheless, the analysis in this paper again stresses the power of analyzing  $O_2$  data in de Fourier domain. As demonstrated here, in cases where the bias is acceptably small, the technique of complex demodulation gives a robust estimate of *daily* GPP in *non-tidal* systems. Although this has also been achieved with classic diel oxygen methods, the crucial advantage here is that no other process has to be estimated. Thus it is not needed to accurately estimate air-water exchange and transport, which is often difficult (Tobias et al., 2009; Kemp and Boynton, 1980; Van de Bogert et al., 2012). However, the biggest
- advantage of Fourier domain analysis lies in *tidal* systems. In those systems, two major reasons make diel oxygen methods difficult to apply 1. the tidal signal dominates  $O_2$  time series and 2. gas transfer is high and difficult to correctly estimate. The Fourier method circumvents these problems to a large extent. First, the M2 component and other semi-diurnals are effectively

filtered out by the analysis. However, as demonstrated above by the analysis of the first order correction term, this does not entirely cancel out the other tidal effects. Indeed, diurnal harmonics in the tide, although much smaller than the semi-diurnals, induce diurnals in  $O_2$  concentrations. When not accounted for, these lead to spurious GPP estimates. One way of accounting for these diurnals is increasing the averaging time to 15 Days. This removes most of the spurious GPP, but also decreases

- the temporal resolution that can be achieved. Second, while air-water transfer of  $O_2$  dampens the fluctuations in  $O_2$ , the resulting underestimation of GPP is small: even with a large piston velocities of  $7ms^{-1}$  and a relatively shallow water depth of about 2.5m, the downward bias when neglecting the dampening is only 10%. The analysis in the frequency domain is not only important for GPP estimation. It enhances theoretical understanding of  $O_2$  dynamics in tidal systems. The mathematical framework of complex demodulation elegantly elucidates how the diurnals  $O_1$  and  $Q_1$  affect  $O_2$  concentrations. They result
- in time varying diurnal O<sub>2</sub> amplitudes, not related to GPP. This explains why the low frequency fluctuations |f1 fO1| and |f1 fQ1| also show up as spurious GPP when classic Odum methods are applied.

### 6 Code and data availability

The Fourier method and complex demodulation described in this manuscript are implemented in a R-package (Cox, 2017). The essential data and scripts to reproduce the analysis in this manuscript are part of the package.

*Author contributions*. Tom J.S. Cox Contributed to theory development, numerical simulations and manuscript writing. Karline Soetaert and Justus van Beusekom contributed to manuscript writing.

*Acknowledgements.* We are grateful to Götz Flöser and Rolf Riethmüller from HZG for providing the Hörnum Tief time series. This work received funding from the EU (grant nr. MARE/2012/10). Tom J.S. Cox thanks Belgian Science Policy for the Belspo Return Grant (selection 2012) he received, which enabled this research.

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

Figure 1. The analogy between sound transmission with AM radio and GPP estimation from O2 time series

Figure 2. Simulated  $O_2$  concentrations with the open water model (top); daily averaged GPP rates (bottom). GPP is reconstructed based on complex demodulation of the simulated  $O_2$  time series.