# Peer review of "Tune in on 11.57 $\mu$ Hz and listen to primary production"

_Biogeosciences, 2017_

## Referee Comment (RC1) · Anonymous Referee #1 · 8 May 2017

This study presents a significant development to the frequency-based GPP estimation method by Cox et al. (2015, doi:10.1002/lom3.10046). The backbone of the method is the same: GPP is extracted from dissolved $O_2$ time-series by assuming that it is the only significant component having an exactly 1 day period. Contrary to the preceding study, the novel method theoretically allows getting daily GPP without having to split the $O_2$ time-series in windows and considering GPP being constant inside the windows, thanks to the complex demodulation technique.

This elegant technique will really contribute to metabolism estimations, but future appliers must know under which circumstances the assumptions remain acceptable. Potential violations of the basic assumption (e.g. that GPP is the only 1-day periodic process within the $O_2$ time-series) are extensively discussed in Cox et al. (2015), but only briefly mentioned here. GPP must change relatively slowly – so water mustn't be eutrophic,

the water column must be deep enough to provide stability but it should not stratify: Shallow systems are not recommended due to intense atmospheric exchange, deep systems due to hindered vertical mixing. Besides the mentioned limitations, others may be identified too: atmospheric exchange can show strong diurnal period when diurnal DO fluctuations are high (e.g. in hypertrophic conditions daily maxima can reach >150% saturation and minima can drop below 2 mg L$^{-1}$ on the same day, generating a huge saturation deficit/surplus) or when piston velocity is affected by regular coastal winds (large lakes often generate systematic local winds at dawn and sunset) and their corresponding horizontal currents.

The paper would benefit from some revision to put more focus on the real benefits of the proposed method. It would be nice if the benchmark could be Cox et al. (2015), and not the bookkeeping method again.

In summary – and in accordance with the paper's statement –, the proposed technique seems to be most suitable to estuary applications, where frequency-domain calculations allow separating the tidal components in the $O_2$ time-series. However, due to the presence of some almost daily tidal components it turns out that a 15 d filter needs to be applied on GPP in order to get a reliable estimate. At this point, it would be nice if the authors could explicitly point out the benefits of the new method compared to the one in Cox et al. (2015), where mean GPP was calculated in 14 d sliding windows, which apparently provided the same results.

Overall, the presentation of results should be revised to really focus on the benefits of the new method, especially when compared to Cox et al. (2015). While this method has theoretically finer temporal resolution, the same kinds of figures are provided (fig 1,2,4top: time vs. estimated and modelled GPP), leaving some doubt it the new method delivers anything more than the previous one. Relations between estimation error and time or $O_2$ are not shown.

Convincing power could be increased if:

- Fig 1 (bottom), 2 (bottom), and 4 (top) would show something more than the corresponding versions in Cox et al. (2015) [fig. 3 top, fig. 6 top].

- the only non-synthetic application (Hörnum Tief) included some reference data, like in Cox et al. (2015). In its present version there is no way to judge if the calculated values had any reference to reality.

SPECIFIC COMMENTS

Pages 5-6: Findings about tidal components should be better organized in results: please use some subsections.

Page 7 Lines 4-5: These Fourier techniques make the implicit assumption that air-water exchange has a period that is far from 1 day, which in practice would mean a constant exchange rate during a day (shorter periods than a day seem unrealistic). This is not conceptually superior to the assumptions made in traditional methods. For the case of transport it may be true (but see remark about coastal winds and corresponding currents above)

Page 8 Lines 3-5: Half of the rather brief conclusions relate to the preceding study. It would be nice to achieve a healthier balance between the new and the old findings. The last sentence isn't really a conclusion: What does 'they' stand for? Moreover, nothing is said about 'HOW they influence' GPP estimates.

TECHNICAL CORRECTIONS

Page 5 Line 31: change "interfe" to "interfere"

Equation 6: in second equation, shouldn't it be y'(t) instead of y(t)?

Page 8 Line 4: change "esatimate" to "estimate"

Figure 1: Please add proper units to the O2 concentration ($\mu$M L$^{-1}$?).

Figure 2: Please add proper units to the O2 concentration ($\mu$M L$^{-1}$?) and GPP ($\mu$M

m$^{-2}$ d$^{-1}$?).

Figure 4: Please add proper units to GPP ($\mu$M m$^{-2}$ d$^{-1}$?) and label plus units to the horizontal axis of the middle panel (f [d$^{-1}$]?). Please add labels to the vertical lines of P1-K1, O1, Q1.

Figure 5: Please add labels to the vertical lines of P1-K1, O1, Q1.

Figure 6: Please add proper units to GPP. Please add labels to the vertical lines of P1-K1, O1, Q1.

---

## Referee Comment (RC2) · Anonymous Referee #2 · 12 Jul 2017

The manuscript show new insights for the in situ calculation of primary production, which is a mile step in marine science. Oxygen measurements are available for many locations and even long time series are available. The background and calculations is quite complicated. I assume that the author would like to address an audience like me, interested and familiar with PP measurements but not too deep into wavelength physics. To address this audience the author need to explain his ambitions more clearly. I wonder where this 11,57 $\mu$HZ wavelength comes from. It is not mentioned in the paper. Some schematic overview figures might help to explain the calculation steps the wave length theory. It would be nice to have a few more background information about tidal consistent O1, Q1, P1 and K1. Most of the figures are poorly labelled, which makes the understanding even harder.

[Figure]

In more detail:

Introduction What does O1 Q1 K1 and P1 stand for?

Where does the idea with the carrier wave come from. Why is it 11,57$\mu$Hz. How is it possible that it is always the same wavelength independent from e.g. the season?

Maybe a schematic overview would help me to understand it. Page 1, line 1: an an

Page 1, line 2: on on

Page 2, line 32: on on

Page 3, line 7: order order

Page 4, line 28: where does the dynamic of the biomass come from?

Results: Fig. 1: Can you add the measured O2 values into top figure og Fig. 1? Please add the unit to GPP. I would call it reconstructed GPP or GPP from complex demodulation. That would be less confusing.

Page 5, line 5 O2

Page 5, line 13: It might come as a surprise... Should be part of the discussion to say how good your method is.

Fig. 2: Include measured O2 data, unit of GPP and use the same names for the legend and color order in the legend.

What is a 1 day low pass filter? Maybe you could already make more clear which method was used for which figure at the end in the material and method part. Maybe it is possible to build up the whole story less in the strict structure of a paper and rather as a story saying we did this, found this which lead us to the next step the use of a 1 day low pass filter.

Page 5, line 22: Most it be overestimation? Isn't it possible that your calculation is better than the simulation?

Fig. 3: What is |F(x)|? It is very hard to find O1 and K1 ... in the figure. Has Fx or f a unit? Please include.

Page 5, line 26: diurnal frequency is f [cycles per day]? Constant nomenclature makes understanding easier. Is it a point for every day of the year? Why are the dots connected with lines?

Does the 11,57 $\mu$ Hz comes from one of the calculated Amplitudes?

Fig. 4: (center) please add labeling of x-axis and unit for F(x).

Page 6, line 1pp. The explanation of the behavior of wavelength is important but I would not include it into the results part.

Page 6, line 7 (T= ...)

What does it mean that the period of the amplitude is 365.1

Fig. 5: (top) again axis label, what does List$WLres mean? What index? Are there any units? (bottom) Units and label please. Where do I see P1, K1? What are the lines? In the description (bottom) is not mentioned. Plus the second sentence should be part of the results it is not a figure description.

The 11,57 $\mu$Hz is not mentioned in the text at all.

Page 7, line 16: O2

––––––––––––––––––––––––––––––

---

## Author Comment (AC1) · 21 Aug 2017

We thank this reviewer for her/his overall supportive comments and suggestions. Below we summarize the comments and suggestions, and we outline how we can incorporate these in a revised manuscript.

Comment: «This study presents a significant development to the frequency-based GPP estimation method by Cox et al. (2015, doi:10.1002/lom3.10046) [...] This elegant technique will really contribute to metabolism estimations, but future appliers must know under which circumstances the assumptions remain acceptable.»

Response: We aim for a minimal overlap between the 2 papers. We have already presented a brief summary these assumptions in the discussion section. But it is a

good suggestion to elaborate a little bit more on this. The discussion of the comparison of our results with some reference data (also suggested by this reviewer, see below) provides a good opportunity.

Comment: «At this point, it would be nice if the authors could explicitly point out the benefits of the new method compared to the one in Cox et al. (2015), where mean GPP was calculated in 14 d sliding windows, which apparently provided the same results. Overall, the presentation of results should be revised to really focus on the benefits of the new method, especially when compared to Cox et al. (2015). While this method has theoretically finer temporal resolution, the same kinds of figures are provided (fig 1,2,4top: time vs. estimated and modelled GPP), leaving some doubt it the new method delivers anything more than the previous one. Relations between estimation error and time or O 2 are not shown.»

Response: There are a number of benefits to the approach presented here. The major benefit is that complex demodulation gives a theoretically consistent framework to deal with time-varying amplitudes, and thus time-varying GPP. This is a major difference with the approach in Cox et al (2015). There, the theoretical derivation relies on O2-fluctuations with constant diurnal amplitude. The pragmatic approach to deal with time varying GPP was to apply the results on moving windows.

As a surprising result of the current approach, we find that (in non-tidal systems) the temporal resolution is very fine: daily values of GPP estimate can be estimated. This would not be possible with the approach of Cox et al 2015.

A second advantage is that this theoretical framework allows to understand and analyze the impact of tidal harmonics. The impact of close to diurnal harmonics on the O2 signal explains why the GPP estimates, when calculated with a 1 day filter are apparently fluctuating (Hoernum Tief site results) with $\sim$ 15 day period. As a result of this theoretically derived impact, we propose an averaging time of 15 days when applying Fourier methods in tidal systems. This result would not be able with the approaches in

Cox et al 2015. We will make this more explicit in a revised version of the paper.

Comment: «Convincing power could be increased if:

• Fig 1 (bottom), 2 (bottom), and 4 (top) would show something more than the corresponding versions in Cox et al. (2015) [fig. 3 top, fig. 6 top]. »

Response: The figures in this paper and the ones in Cox et al 2015 are fundamentally different. Figures 1 and 2 here show daily GPP estimates, while in Cox et al 2015, 10day moving average of GPP was calculated. Figure 4 top shows the impact of a first order correction term on the estimate, demonstrating that this first order term is the major cause of the mismatch between the true GPP and 1 daily GPP estimate by complex demodulating the O2 time series. Nevertheless, the simulation on which these calculations are based are performed with the same model, hence the resemblance of the figures. We will clarify this in a revised version.

Comment: «• the only non-synthetic application (Hörnum Tief) included some reference data, like in Cox et al. (2015). In its present version there is no way to judge if the calculated values had any reference to reality. »

Response: We thank this reviewer for this suggestion. Reference data from a nearby site can be added to a revised version of the manuscript, this will simultaneously allow us to briefly discuss the assumptions underlying the Fourier methods (see above).

Comment: «Pages 5-6: Findings about tidal components should be better organized in results: please use some subsections.»

Response: OK

Comment: «Page 7 Lines 4-5: These Fourier techniques make the implicit assumption that air-water exchange has a period that is far from 1 day, which in practice would mean a constant exchange rate during a day (shorter periods than a day seem unrealistic). This is not conceptually superior to the assumptions made in traditional methods. For the case of transport it may be true (but see remark about coastal winds and corresponding currents above)»

Response: Diel fluctuations in air water exchange are indeed assumed to be small compared to diel fluctuations due to GPP. This is part of the assumptions which we will elaborate more on in a revised version of the MS.

Comment «Page 8 Lines 3-5: Half of the rather brief conclusions relate to the preceding study. It would be nice to achieve a healthier balance between the new and the old findings.»

Response This section will be revised to achieve a healthier balance between the new and the old findings.

---

## Author Response (AR1)

Dear,

Below the detailed author's response to the reviewer comments is appended.

The whole manuscript was revised and rewritten to meet the comments and suggestions by the reviewers.

– The introduction was rewritten to put less focus on the bookkeeping method.
– A conceptual figure is added to help understanding of the basic concept
– Reference data is added. This includes field data of bottle incubations in 2004 and Chl a data to assess the comparison. This is incorporated throughout methods, results and discussion sections
– To improve readability, results section is rewritten and clarified, and has been structured in different subsection with specific subtitles.
– Discussion section is extended with a focus on the benefits of the new method compared to earlier results. It also incorporates a discussion of the basic asssumptions behind Fourier methods, in relation to the results on field data.
– Scripts and software to perform complex demodulation are now available as an R-package on CRAN.
– Additionally:
Figures have been improved,
– axis labels added where previously omitted or incomplete,
– additional, clarifying information on the figures
– colors and legends made consistent
As suggested by the reviewers we now talk of "Simulated GPP" and "Reconstructed GPP" throughout.

We again thank the reviewers for their valuable comments. Incorporating their comments and suggestion in our revision has significantly improved the manuscript.

We look forward to the decision,

On behalf of all co-authors
Tom Cox
University of Antwerp

—--------------------

**Response to comments of reviewer 1**

**We thank this reviewer for her/his overall supportive comments and suggestions. Below we summarize the comments and suggestions, and we outline how we can incorporate these in a revised manuscript.**

<<This study presents a significant development to the frequency-based GPP estimation method by Cox et al. (2015, doi:10.1002/lom3.10046) [...] This elegant technique will really contribute to metabolism estimations, but future appliers must know under which circumstances the assumptions remain acceptable.>>

**We aim for a minimal overlap between the 2 papers. We have already presented a brief summary these assumptions in the discussion section. But it is a good suggestion to elaborate a**

**little bit more on this. The discussion of the comparison of our results with some reference data (also suggested by this reviewer, see below) provides a good opportunity.**

<<At this point, it would be nice if the authors could explicitly point out the benefits of the new method compared to the one in Cox et al. (2015), where mean GPP was calculated in 14 d sliding windows, which apparently provided the same results. Overall, the presentation of results should be revised to really focus on the benefits of the new method, especially when compared to Cox et al. (2015). While this method has theoretically finer temporal resolution, the same kinds of figures are provided (fig 1,2,4top: time vs. estimated and modelled GPP), leaving some doubt it the new method delivers anything more than the previous one. Relations between estimation error and time or O 2 are not shown.>>

**There are a number of benefits to the approach presented here. The major benefit is that complex demodulation gives a theoretically consistent framework to deal with time-varying amplitudes, and thus time-varying GPP. This is a major difference with the approach in Cox et al (2015). There, the theoretical derivation relies on $O_2$-fluctuations with constant diurnal amplitude. The pragmatic approach to deal with time varying GPP was to apply the results on moving windows. In Cox et al (2015) we were not able to**

**As a surprising result of the current approach, we find that (in non-tidal systems) the temporal resolution is very fine: daily values of GPP estimate can be estimated. This would not be possible with the approach of Cox et al 2015.**

**A second advantage is that this theoretical framework allows to understand and analyse the impact of tidal harmonics. The impact of close to diurnal harmonics on the $O_2$ signal explains why the GPP estimates, when calculated with a 1 day filter are apparently fluctuating (Hoernum Tief site results) with ~ 15 day period. As a result of this theoretically derived impact, we propose an averaging time of 15 days when applying Fourier methods in tidal systems. This result would not be able with the approaches in Cox et al 2015. We will make this more explicit in a reved version of the paper.**

<<Convincing power could be increased if:

• Fig 1 (bottom), 2 (bottom), and 4 (top) would show something more than the corresponding versions in Cox et al. (2015) [fig. 3 top, fig. 6 top]. >>

**The figures in this paper and the ones in Cox et al 2015 are fundamentally different. Figures 1 and 2 here show daily GPP estimates, while in Cox et al 2015, 10day moving average of GPP was calculated. Figure 4 top shows the impact of a first order correction term on the estimate, demonstrating that this first order term is the major cause of the mismatch between the true GPP and 1 daily GPP estimate by complex demodulating the $O_2$ time series. Nevertheless, the simulation on which these calculations are based are performed with the same model, hence the resemblance of the figures. We will clarify this in a revised version.**

<<• the only non-synthetic application (Hörnum Tief) included some reference data,
like in Cox et al. (2015). In its present version there is no way to judge if the
calculated values had any reference to reality. >>

**We thank this reviewer for this suggestion. Reference data from a nearby site can be added to a revised version of the manuscript, this will simultaneously allow us to briefly discuss the assumptions underlying the Fourier methods (see above).**

<<Pages 5-6: Findings about tidal components should be better organized in results: please use some subsections.>>
**OK**

<<Page 7 Lines 4-5: These Fourier techniques make the implicit assumption that air-water exchange has a period that is far from 1 day, which in practice would mean a constant exchange rate during a day (shorter periods than a day seem unrealistic). This is not conceptually superior to the assumptions made in traditional methods. For the case of transport it may be true (but see remark about coastal winds and corresponding currents above)>>
**Diel fluctuations in air water exchange are indeed assumed to be small compared to diel fluctuations due to GPP. This is part of the assumptions which we will elaborate more on in a revised version of the MS.**

<<Page 8 Lines 3-5: Half of the rather brief conclusions relate to the preceding study. It would be nice to achieve a healthier balance between the new and the old findings.>>

**This section will be revised to achieve a healthier balance between the new and the old findings.**

_----------------------_
**Response to comments of reviewer 2**

**We thank this reviewer for her/his overall supportive comments and suggestions. Below we summarize the comments and suggestions, and we outline how we can incorporate these in a revised manuscript.**

Comment:
<<The manuscript show new insights for the in situ calculation of primary production, which is a mile step in marine science. [...] The background and calculations is quite complicated. I assume that the author would like to address an audience like me, interested and familiar with PP measurements but not too deep into wave-length physics. To address this audience the author need to explain his ambitions more clearly. I wonder where this 11,57 µHZ wavelength comes from. It is not mentioned in the paper. Some schematic overview figures might help to explain the calculation steps the wave length theory. It would be nice to have a few more background information about tidal consistent O1, Q1, P1 and K1. Most of the figures are poorly labelled, which makes the understanding even harder.>>

Response
**1 cycle / day = 1 cycle per 86400 seconds = 11.57 micro Hz. (See also L36 of the introduction). A sketch will indeed clarify the concepts of the carrier wave with frequency 11.57 micro Hz for many readers. We will add it to a revised manuscript.**

**Tides are known to be a superposition of different periodic functions with well known frequencies. O1, Q1, P1 and K1 are the components of the tides, with close to diurnal frequency (See L27 of the Results). We will clarify this in a revised manuscript.**

**We will revise the labels of the figures.**

Comment

<<In more detail:
Introduction What does O1 Q1 K1 and P1 stand for?...>>

**Response:**
**See above**

Comment:
<<Page 4, line 28: where does the dynamic of the biomass come from?>>

Response:
**A season of primary production is dynamically simulated, with a seasonal build-up and break-down of algal biomass . We will clarify this short section**

**Comment:**
<<Results: Fig. 1: Can you add the measured O2 values into top figure og Fig. 1?
Please add the unit to GPP. I would call it reconstructed GPP or GPP from complex demodulation. That would be less confusing.>>

Response:
**We will do so in a revised version**

Comment:
<<What is a 1 day low pass filter? Maybe you could already make more clear which method was used for which figure at the end in the material and method part. Maybe it is possible to build up the whole story less in the strict structure of a paper and rather as a story saying we did this, found this which lead us to the next step the use of a 1 day low pass filter.>>

Response:
Here, we use a moving average filter with a width of 1 day. We will clarify this in a revised manuscript.

Comment:
<<Page 5, line 22: Most it be overestimation? Isn't it possible that your calculation is better than the simulation?>>

Response:
**No, if the method would be perfect, the simulated GPP would be perfectly reproduced.**

**Comment:**
<<Fig. 3: What is |F(x)|? It is very hard to find O1 and K1 . . . in the figure. Has Fx or f a unit? Please include.>>

Response:
**We will clarify this**

Comment:
<<Page 6, line 7 (T= . . .)
What does it mean that the period of the amplitude is 365.1>>

**The text literally says: "the period of the amplitude *variation* is 365.1 days". The superposition of K1 and P1 tidal harmonics result in a signal with frequency very close to the diel frequency. Whereas the frequency of this signal is constant (diurnal), the amplitude varies periodically. The period of this amplitude variation is 365.1 days. We will clarify this in a revised version.**

<<Fig. 5: (top) again axis label, what does List$WLres mean? What index? Are there any units? (bottom) Units and label please. Where do I see P1, K1? What are the lines? In the description (bottom) is not mentioned. Plus the second sentence should be part of the results it is not a figure description.>>
**We will revise the figures**

Comment:
<<The 11,57 μHz is not mentioned in the text at all.>>
**It is mentioned in the text, but we will add a sketch to clarify, see above.**